RESEARCH

# Decoupling of the minority PhD talent pool and assistant professor hiring in medical school basic science departments in the US

**Abstract** Faculty diversity is a longstanding challenge in the US. However, we lack a quantitative and systemic understanding of how the career transitions into assistant professor positions of PhD scientists from underrepresented minority (URM) and well-represented (WR) racial/ethnic backgrounds compare. Between 1980 and 2013, the number of PhD graduates from URM backgrounds increased by a factor of 9.3, compared with a 2.6-fold increase in the number of PhD graduates from WR groups. However, the number of scientists from URM backgrounds hired as assistant professors in medical school basic science departments was not related to the number of potential candidates ($R^2$=0.12, p>0.07), whereas there was a strong correlation between these two numbers for scientists from WR backgrounds ($R^2$=0.48, p<0.0001). We built and validated a conceptual system dynamics model based on these data that explained 79% of the variance in the hiring of assistant professors and posited no hiring discrimination. Simulations show that, given current transition rates of scientists from URM backgrounds to faculty positions, faculty diversity would not increase significantly through the year 2080 even in the context of an exponential growth in the population of PhD graduates from URM backgrounds, or significant increases in the number of faculty positions. Instead, the simulations showed that diversity increased as more postdoctoral candidates from URM backgrounds transitioned onto the market and were hired.

**KENNETH D  GIBBS JR**[*]**, JACOB BASSON, IMAM M  XIERALI AND DAVID A BRONIATOWSKI**

*For correspondence: kgibbsjr@gmail.com

## Introduction

Enhancing the diversity of the research workforce has been a longstanding priority of scientific funding agencies (*Tabak and Collins, 2011*; *Valantine and Collins, 2015*; *National Institutes of Health, 2015*; *National Institute of General Medical Sciences, 2015*). Scientists from certain underrepresented minority (URM) racial/ethnic backgrounds—specifically, African American/Black, Hispanic/Latin@, American Indian, and Alaska Native—receive 6% of NIH research project grants (*Ginther et al., 2016*,

*2011*; *National Institutes of Health, 2012b*) despite having higher representation in the relevant labor market (*Heggeness et al., 2016*), and constituting 32% of the US population (*National Institutes of Health, 2012b*). The vast majority of NIH funding—approximately 83%—is awarded to investigators at extramural institutions, many of whom serve as faculty members at academic and research institutions (*Johnson, 2013*). In particular, MD-granting medical schools and their affiliates (henceforth, medical schools) that belong to the Association of American Medical Colleges (AAMC) receive 67% of

NIH extramural funding, and comprised the entire top 20 of NIH-funded institutions in FY2015 (*National Institutes of Health, 2016*). As a result, the goal of diversifying the biomedical investigator pool necessitates diversifying the professoriate generally, and in medical schools specifically.

Faculty members play critical and unique roles within the scientific enterprise, shaping the national research agenda, and cultivating the next generation of scientists and scholars (*Clauset et al., 2015*; *Leggon, 2010*). However, a 2011 report from the National Academies of Sciences said "diversifying faculties is perhaps the least successful of the diversity initiatives" (*National Academy of Sciences, 2011*). Student protests on college campuses across the country in the 2015 academic year often centered on the need for more faculty diversity, and highlighted the lack thereof, especially in scientific disciplines (*Griffin, 2016*). As the nation continues to diversify, broadening participation within the research enterprise and professoriate is believed to be critically important for maintaining an adequate domestic scientific workforce, and ensuring the research enterprise effectively meets the needs of the entire population (*National Institutes of Health, 2012b*; *National Academy of Sciences, 2011*).

This work focuses on three possible reasons for the low number of scientists from URM backgrounds in the professoriate relative to their peers from well-represented (WR) backgrounds (specifically, White, Asian, and all other non-URM groups) that are amenable to intervention by the scientific community: (i) the size of the URM PhD talent pool, (ii) the number of available faculty positions, and (iii) the transition of the available URM PhD and postdoctoral talent pool onto the faculty job market, and their subsequent hiring. Educational disparities between students from URM and WR backgrounds begin early in life, and accumulate from K-12 through early independence (*National Institutes of Health, 2012b*; *Garrison, 2013*). Thus, it is possible that the cumulative impact of these disparities is the URM PhD and postdoctoral talent pool that is too small to sustain meaningful levels of faculty diversity (*Garrison et al., 2009*). If so, intervention strategies would need to focus primarily on building the talent pool.

Additionally, current faculty diversity efforts occur against the backdrop of systemic changes within biomedicine. Following the doubling of the NIH budget between 1998 and 2003, there was a significant increase in the number of PhDs awarded, without commensurate increase in the number of faculty positions (*Stephan, 2012*; *Alberts et al., 2014*). This led to labor market imbalances in which there are significantly more scientists who desire faculty positions than the supply of such positions. Further, it is estimated that fewer than 11% of all life science PhDs enter faculty positions in any institution type (*National Science Board, 2014*). This raises the possibility that the low number of faculty from URM groups is mainly a function of broader stresses on the faculty job market or changes in the overall labor market for PhDs (*Zolas et al., 2015*). If so, intervention strategies could focus on expanding the number of new faculty positions available, thus creating more opportunities for scientists from all backgrounds.

Beyond the number of faculty positions available, there is evidence that graduate students and postdocs from all backgrounds lose interest in faculty careers in research-intensive universities as their training progresses (*Fuhrmann et al., 2011*; *Gibbs et al., 2015*; *Sauermann and Roach, 2012*). Moreover, at PhD completion URM men and women report lower levels of interest in faculty positions at research-intensive universities than their WR counterparts, even when controlling for career interests at PhD entry, scholarly productivity, mentorship or research self-efficacy (*Gibbs et al., 2014*). Thus, part of the lack of representation could be due to disproportionately low application rates by URM PhD graduates and postdocs for these positions for reasons ranging from values misalignment (*Gibbs et al., 2013*), implicit and explicit biases (*Colon Ramos and Quiñones-Hinojosa, 2016*; *Jarvis, 2015*), or perceptions of hypercompetition within academic research that makes the positions particularly unattractive in the current funding climate (*McDowell et al., 2014*).

Increasing diversity in the applicant pool and equitable evaluation in the hiring process are strategies that promote faculty diversity (*Turner, 2002*; *Sheridan et al., 2010*; *Moody, 2004*; *Smith, 2015*). While systematic data are not available on the demographics of faculty applicants, the chair of a recent faculty search in systems biology at Harvard university reported very low numbers of applications from women and scientists from URM backgrounds (*Eddy, 2015*), lending credence to the notion that faculty applicant pools lack diversity. If this is the case, intervention strategies could focus on enhancing diversity in the applicant pool and

ensuring equitable evaluation to increase faculty diversity.

The static nature of faculty diversity, especially in research-intensive environments, suggests that new approaches are necessary for achieving the goal of workforce diversity. In particular, computational modeling approaches such as System Dynamics (SD) have been used to examine the macro-scale impacts of potential policy interventions on the biomedical postdoctoral workforce (*Ghaffarzadegan et al., 2014*), and new faculty hiring (*Larson and Diaz, 2012*). The goal of this work is to:

1. Provide a systematic and quantitative perspective on changes in the numbers of biomedical PhDs and assistant professorships in medical school basic science departments by scientists from URM and WR backgrounds between 1980-2014.
2. Build and validate a System Dynamics (SD) model that can capture major trends in the career progression of PhD scientists from URM and WR backgrounds into this segment of the professoriate.
3. Utilize the SD model to test the impact of various intervention strategies to faculty diversity in the short-term (through 2030) and long-term (through 2080). Specifically, we model the impact on faculty diversity at the assistant professor stage by increasing: (i) the size of the talent pool of PhDs from URM backgrounds, (ii) the number of assistant professor positions available, or (iii) the rate of transition of PhDs from URM backgrounds into the applicant pool of assistant professorships.

We focus on medical school basic science departments because of the availability of comprehensive, longitudinal demographic data (in contrast to the broader biomedical workforce, where career outcome data are lacking [*Polka et al., 2015*; *National Institutes of Health, 2012a*]). Our goal is that these analyses can provide an example for other areas of the scientific community working to address their own diversity challenges.

## Results

### Trends in PhD Graduation and assistant professorship growth: 1980-2014

*Figure 1* shows how the representation of scientists from URM and WR backgrounds in the populations of biomedical PhD graduates, and assistant professors in medical school basic science departments has changed from 1980-2014

(complete data are available in *Figure 1—source data 1*). These analyses include the annual population (*Figure 1Ai,Bi*), population growth relative to 1980 (*Figure Aii,Bii*), and the percentages of scientists from each population from each group (*Figure 1Aiii,Biii*). Data on the populations of PhD graduates and assistant professors in medical school basic science departments were obtained from the National Science Foundation Survey of Earned Doctorates (as compiled by Federation of American Societies for Experimental Biology), and the AAMC Faculty Roster, respectively (please see methods section for more information).

For both URM and WR populations, there was significant growth in the number of PhD graduates, and significant yet slower growth in the population of assistant professors (*Figure 1*). However, there were differences in the magnitudes of these changes across time. The annual number of URM PhD graduates grew more than nine-fold from 1980–2013 (from n=93 to n=868), whereas the population of URM assistant professors grew 2.6-fold (from n=132 in 1980 to n=341 in 2014; *Figure 1Ai–ii*). In comparison, for scientists from WR backgrounds growth in assistant professors was more closely aligned with growth in PhD graduates–there was a 2.2-fold increase in the annual number of PhD graduates (from n=3989 in 1980 to n=8789 in 2013; *Figure 1Bi–ii*), and a 1.7-fold increase for population of assistant professors (n=3246 in 1980 to n=5562 in 2014; *Figure 1Bi–ii*). While the population of PhD graduates grew more quickly than that of assistant professors for all groups over time, this difference was greater in the URM population than the WR population. That is, there was a statistically significant interaction between URM status and position ($\beta=1.60$; $p=3.6*10^{-7}$; PhD graduates relative to assistant professors), above the impacts URM status ($\beta=0.0602$, $p=0.005$), position alone ($\beta = 0.229$, $p=0.28$), or the increases that occurred as the system grew through time ($\beta = 0.0895$, $p=2*10^{-16}$).

*Figure 1Aiii and Biii* show the proportions of URM and WR PhD graduates in the overall pool (solid lines) and among U.S. citizens and permanent residents (dotted lines). Among the pool of U.S. citizens and permanent residents, the proportion of URM PhD graduates grew from 2.5% in 1980 to 13% in 2013, whereas in the overall pool the proportion of URM PhD graduates grew from 2.3% in 1980 to 9% in 2013. In contrast, the percentage of URM assistant professors grew from 3.9% in 1980 to 5.8% in 2014 (*Figure 1Aiii*).

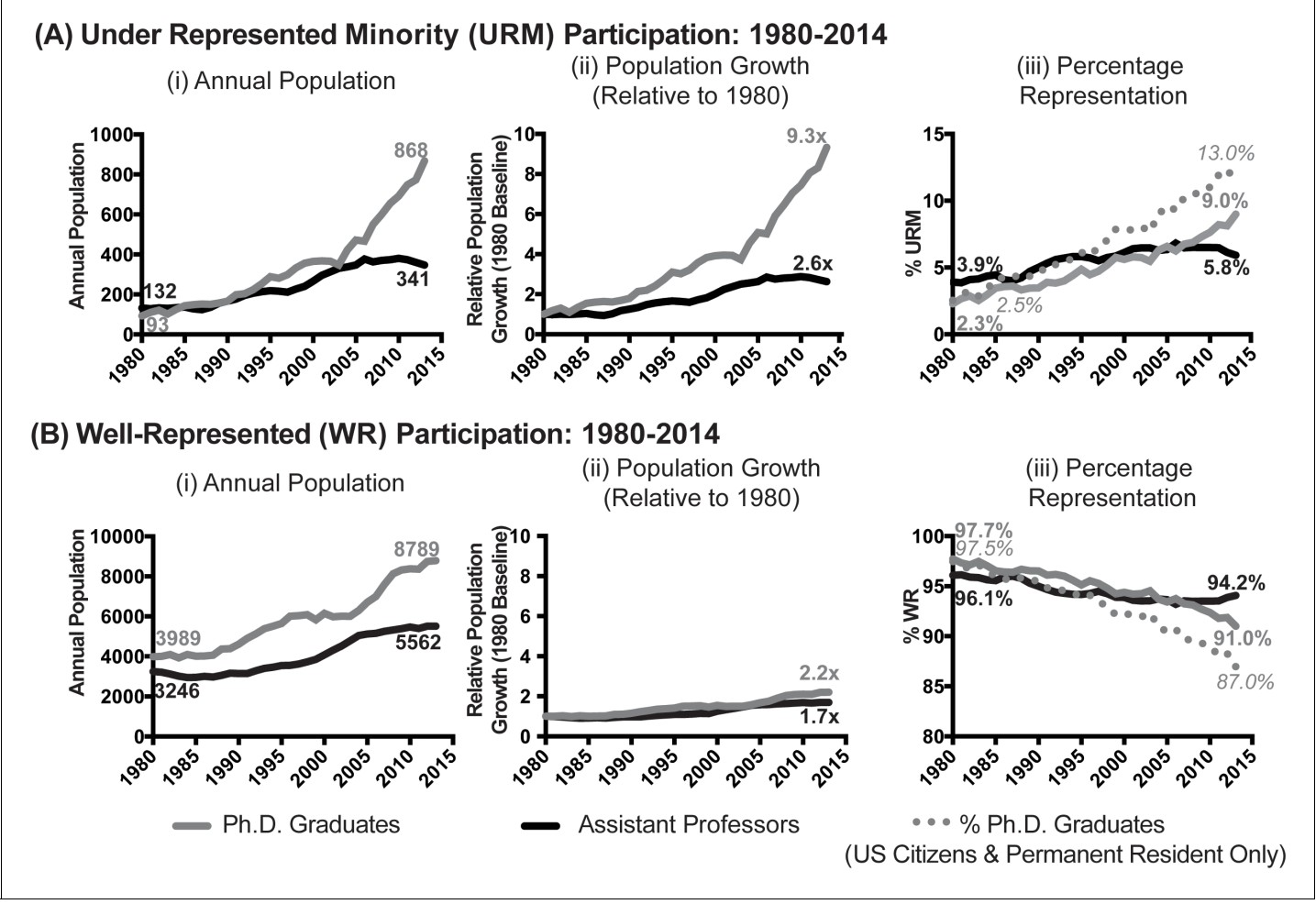

**Figure 1.** Temporal trends in the populations of biomedical Underrepresented Minority (URM) and Well-Represented (WR) PhD graduates and assistant professors, 1980-2014. Line charts showing the (i) annual population, (ii) population growth relative to 1980, and (iii) percentage representation of PhD graduates and assistant professors in basic science departments in medical schools for scientists from (A) URM and (B) WR racial-ethnic backgrounds. Data on the populations of PhD graduates and assistant professors in medical school basic science departments were obtained from the National Science Foundation Survey of Earned Doctorates (as compiled by Federation of American Societies for Experimental Biology), and the AAMC Faculty Roster, respectively (please see methods section for more information). Grey lines represent PhD graduates, and black lines represent assistant professors. In panels Aiii and Biii, solid grey lines represent the percentages of URM and WR PhD graduates among all students who receive PhDs in the U.S. (U.S. citizen, permanent resident, and international), and dotted lines show percentages among PhD graduates who are U.S. citizens and permanent residents. The relative growth of PhD graduates from URM backgrounds to assistant professors is greater than the same comparison among scientists from WR backgrounds (i.e., there was a significant interaction between the URM status and position, $\beta=1.60$; $p=3.6*10^{-7}$; panels Aii and Bii). Data are available in *Figure 1—source data 1*.

The following source data is available for figure 1:

**Source data 1.** PhD graduates and assistant professors (Total, URM and WR): 1980-2014.

Between 2005-2013, a total of 5,842 biomedical PhDs were awarded to scientists from URM backgrounds; however, there were six *fewer* URM assistant professors in basic science departments in 2014 than in 2005 (n=341 in 2014 versus 347 in 2005). For scientists from WR backgrounds, there was 31% growth in the annual number of PhD graduates (n=8789 in 2013 compared to n=6703 in 2005) and 8.6% growth in the population of assistant professors (n=5562 in 2014 compared to n=5122 in 2005). Thus, while the populations of PhD graduates and assistant professors has grown since 1980 for scientists from all backgrounds, the

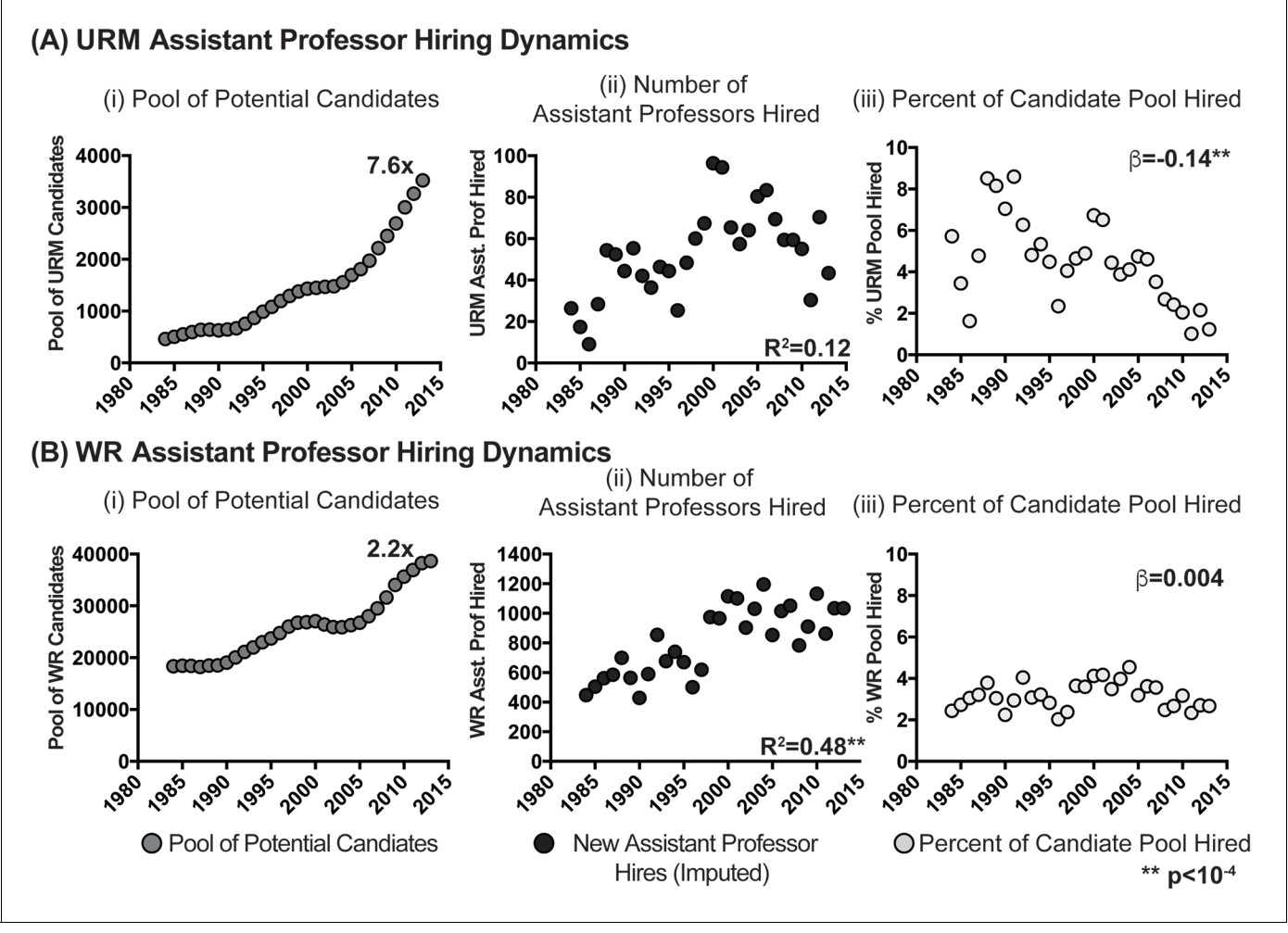

**Figure 2.** Candidate pool size, hiring and utilization of URM and WR assistant professors in basic biomedical science departments. Scatter plots showing the (i) pool of potential candidates for assistant professor positions, (ii) annual number of assistant professors hired, and (iii) percentage of the potential candidate pool hired annually for scientists from (**A**) URM and (**B**) WR backgrounds. $R^2$ values in panels Aii and Bii are derived from correlating number of URM or WR assistant professors hired with the size of their respective pool of potential candidates. β in panels Aiii and Biii reflect the yearly percentage change in the fraction of the pools of URM and WR scientists hired into assistant professor positions. Asterisks represent significant values ($p<10^{-4}$). Data are available in *Figure 2—source data 1* and *2*.

The following source data and figure supplement are available for figure 2:

**Source data 1.** Assistant professor hiring and leaving (total, URM and WR): 1980-2014.

**Source data 2.** Candidate pool and fraction hired (URM and WR): 1980-2014.

**Figure supplement 1.** Candidate pool size, hiring and utilization of URM and WR assistant professors in basic biomedical science departments: by gender.

magnitude of the growth of PhD graduates relative to assistant professors differed greatly between URM and WR scientists.

### Hiring patterns of URM and WR assistant professors in basic science

The patterns of assistant professor hiring differed across populations. For scientists from URM backgrounds, there was a 7.6-fold increase in the size of the potential candidate pool (*Figure 2Ai*); however, the size of the potential URM candidate pool was not significantly correlated with the number of URM assistant professors hired each year ($R^2$=0.12, p=0.07; *Figure 2Aii*). In contrast, for scientists from WR backgrounds there was a 2.2-fold growth in the

size of the candidate pool (*Figure 2Bi*), and the size of the potential candidate pool was significantly correlated with the number of assistant professors hired ($R^2$=0.48, p=2.54*10$^{-5}$; *Figure 2Bii*). For scientists from URM backgrounds, the proportion of the candidate pool hired into assistant professor positions decreased from year to year (β=-0.14, p=9.6*10$^{-4}$), while for scientists from WR backgrounds the proportion of the potential candidate pool hired did not change significantly over time (β=0.004, p=0.77). Thus, despite growth in the pools of potential URM and WR candidates, the nature of entry into assistant professor positions differed significantly between the two populations, with little connection between the size of the URM available candidate pool, and the numbers entering into assistant professor positions (full data are available in *Figure 2—source data 1* and *2*).

### System dynamics model development and calibration

We created a System Dynamics model capturing the flows of PhD graduates from URM and WR backgrounds into assistant professor positions. This abstract model (*Gilbert, 2007*) expands on the traditional "pipeline" view of assistant professor hiring (*Figure 3A*), and is calibrated with the empirical data mentioned above (*Figure 3B* for intermediate conceptual model, and *Figure 3C* for final model; the source code provides the model software file). Hiring trends (e.g. growth in pool size, relationship between potential candidate pool and number of assistant professors hired) were largely consistent across the intersections of gender and URM or WR status (*Figure 2—figure supplement 1*). Thus, for modeling, we focused only on URM/WR status, and not their intersections with gender.

The core assumptions of the model are that the number of assistant professors hired is based on: 1) the number of positions available, and 2) the number of candidates pursuing these positions. Candidates on the market are composed primarily of the subset of postdoctoral scientists pursuing faculty careers in medical school basic science departments (evidence suggests that the rates of transition into postdoctoral training are comparable between URM and WR PhD graduates [*National Science Foundation, 2015b*]). Based on market hiring conditions, one-fifth of the candidates on the market drop out of the market annually. That is, if the probability of being hired is relatively high (>20%) all

candidates remain on the market that year; otherwise, one-fifth of the available pool drops out of the system. Thus, the average "half-life" of a candidate on the market who is not hired is five years (similar to the period of postdoctoral training for candidates pursuing faculty positions in research-intensive environments [*National Institutes of Health, 2012a*]).

The model also posits that URM and WR candidates are hired in direct proportion to their representation on the market. That is, the model's output assumes that that racial bias does not impact hiring. Further, the model assumes that differences in relative strength on the job market across URM status do not impact hiring, because URM PhD graduates have lower interest in faculty careers in research-intensive environments than WR PhDs graduates even if they have graduated from the same institutions and have the same levels of scholarly productivity (*Gibbs et al., 2015*; *Gibbs et al., 2014*).

Based on the analyses presented above, the career pathways for scientists from URM and WR backgrounds were represented separately, but were linked based on the total number of assistant professor slots available. Within each population, we assumed a fixed proportion of graduates would pursue and enter faculty positions in research-intensive environments (i.e. "faculty aspire"). The size of the "faculty aspire" pool was based on hiring trends 1980-1997, before the NIH budget doubling and subsequent expansion of the biomedical PhD pool. All other PhDs would pursue other careers (i.e. "other aspire"). Without intervention, the "faculty aspire" and "other aspire" populations grow in proportion to the total number of PhD graduates (i.e. "baseline PhD graduate growth rate"). We further assumed that efforts by the scientific community to enhance the diversity of the PhD pool ("URM target growth rate") would increase the pool of URM "other aspire" PhDs, some of whom will then transition to the faculty market. Key variables—including baseline PhD graduate growth rate, URM target growth rate, proportions of URM or WR scientists pursuing faculty positions, and the number of positions available—were derived from national survey data, while the transition rate represented a free parameter for analysis. Full details of the model are provided in the methods section and model equations and parameter values are presented in Appendix-Tables 1 and 2.

We calibrated our model against empirical trends in PhD graduations ($R^2$=0.99, p<0.0001;

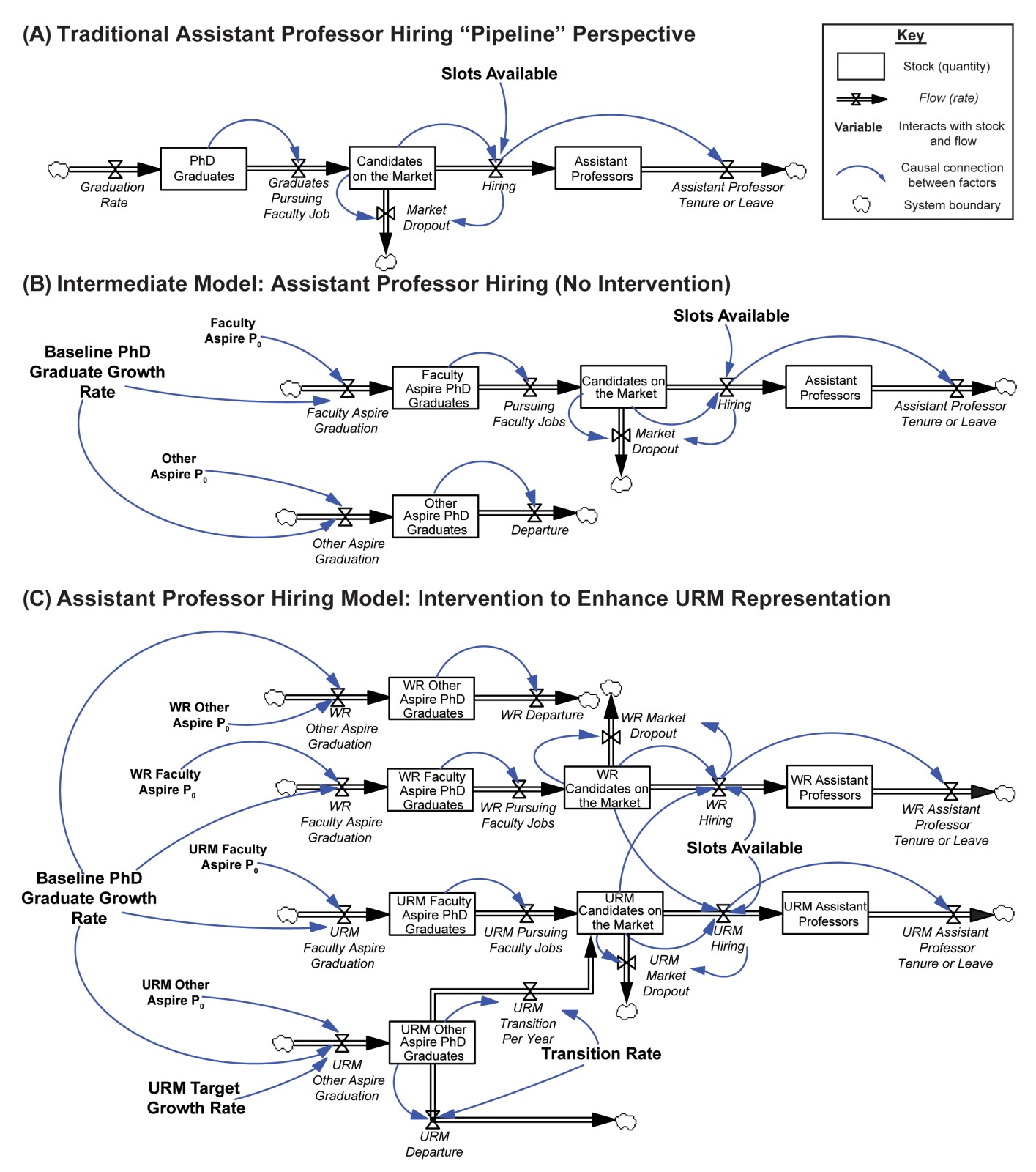

**Figure 3.** System dynamics model of assistant professor hiring. (**A**) A traditional "pipeline" view of faculty hiring. A fraction of the total stock of PhD graduates pursues faculty positions, and thus become candidates on the market. Candidates on the market are composed primarily of the subset of postdoctoral scientists pursuing faculty careers in medical school basic science departments but can include those who have non-traditional career paths such as the rare PhD student who proceeds directly to the faculty job market. Each year, candidates on the market are hired into the stock of

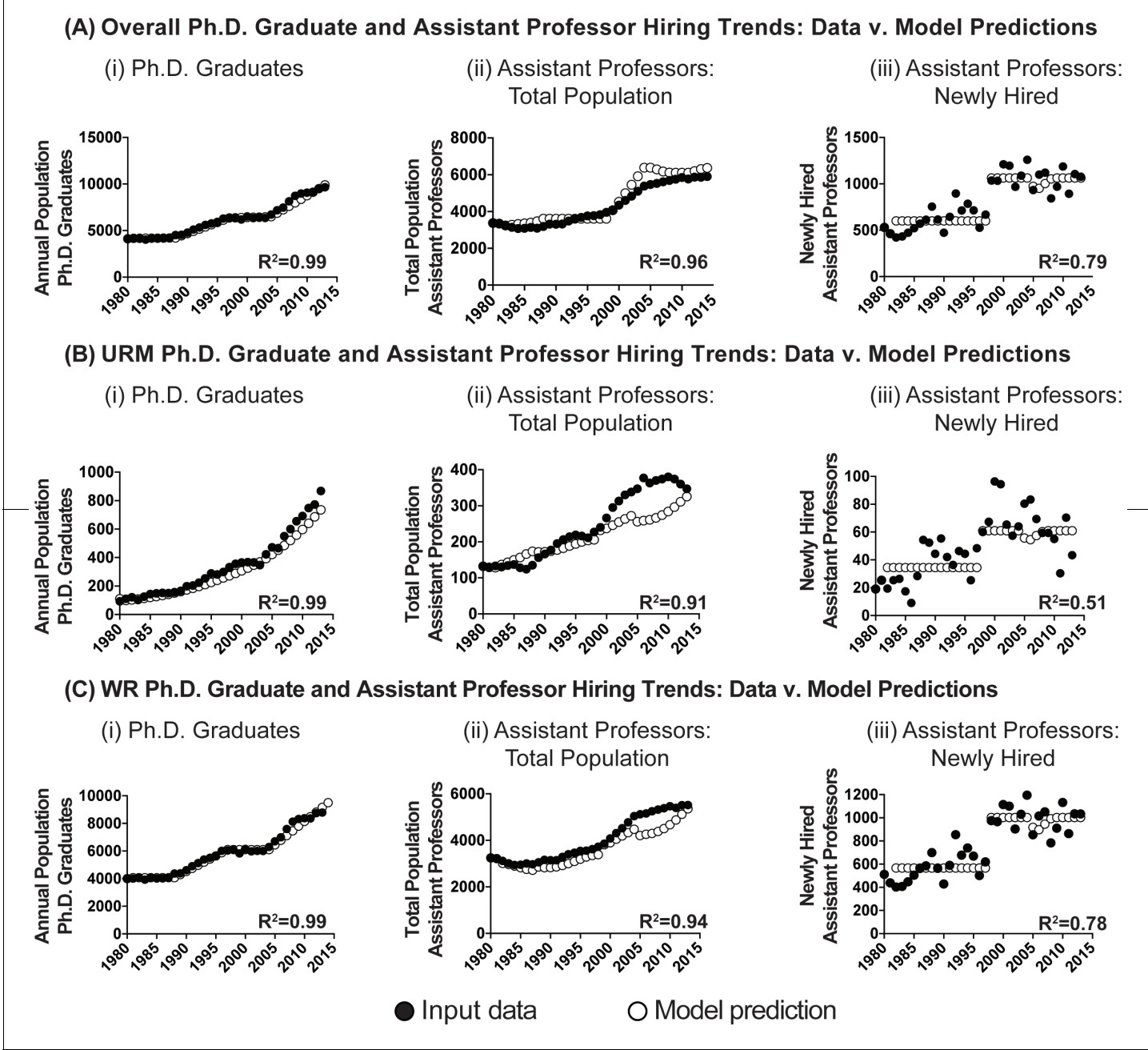

**Figure 4.** Model simulation: 1980-2013. Scatter plots showing the performance of the model (open circles) compared to input data (filled circles) for the populations of (i) PhD graduates, (ii) assistant professors, and (iii) newly hired assistant professors for the (**A**) overall pool, (**B**) pool of URM scientists, and (**C**) pool of WR scientists. All $R^2$ values are significant at the $p<0.0001$ level.

*Figure 4Ai*) and chose the number of available slots to match empirical assistant professor hiring trends ($R^2=0.96$, $p<0.0001$; *Figure 4Aii*). The resulting model output captured 79% of the variance in overall assistant professor hiring ($R^2=0.79$, $p<0.0001$; *Figure 4Aiii*). When disaggregated by URM and WR status, the model captures 51% of the variance in URM hiring (*Figure 4Biii*; $R^2=0.51$, $p<0.0001$), and 78% of the variance in WR assistant professor hiring ($R^2=0.78$, $p<0.0001$; *Figure 4Ciii*). Thus, the model captures major trends in URM and WR hiring rates, over and above what is captured by just examining the size of the talent pool (*Figure 2B*).

## (A) Short-term simulation (through 2030)   (B) Long-term simulation (through 2080)

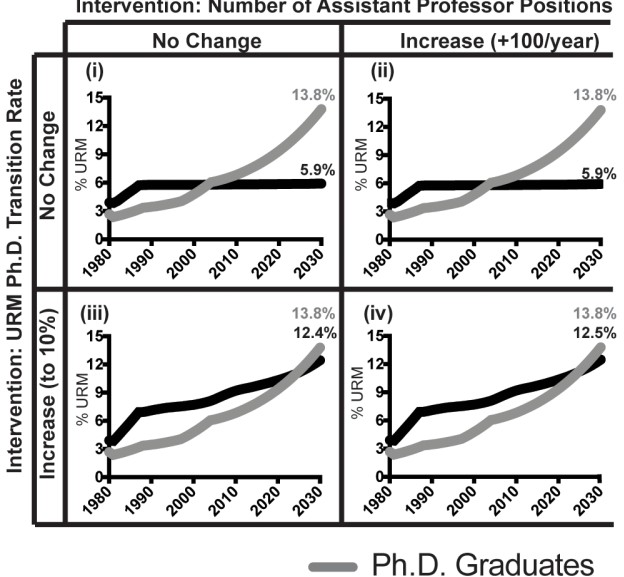
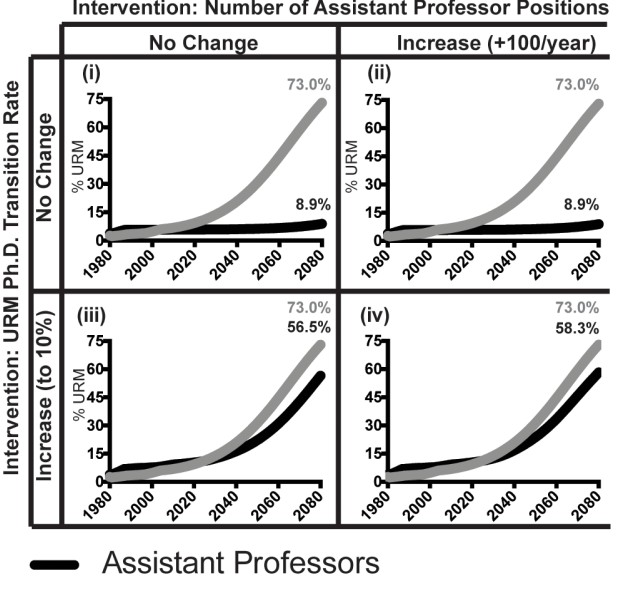

**Figure 5.** Model predictions of URM assistant professor attainment. Line graph showing model predictions for the percentage of URM PhD graduates (grey), and the corresponding percentages of URM assistant professors (black) as a function of various intervention strategies to increase faculty diversity in (A) short-term, through 2030, and (B) long-term, through 2080. All model runs assume an exponential increase in the number of PhDs from URM backgrounds. Thus, in all runs, the percentage of PhD scientists from URM backgrounds is 13.8% in 2030 and 73% in 2080. Simulations: (i) No change in transition rate (0.25%) or number of assistant professor positions. (ii) No change in transition rate (0.25%), increase the number of assistant professor positions by 100 per year, beginning in 2015. (iii) Increase transition rate to 10%, and no change in the number of assistant professor positions. (iv) Increase transition rate to 10% and increase the number of assistant professor positions by 100 per year, beginning in 2015.

The following source data is available for figure 5:

**Source data 1.** Model predictions: percentage URM assistant professors by transition rate: 1980-2080 (current number of assistant professor positions)
**Source data 2.** Model predictions: percentage URM assistant professors by transition rate: 1980-2080 (100 new assistant professor positions, annually, beginning in 2015)

### Intervention strategies to increasing assistant professor diversity

We used the model to test the impact of three different intervention strategies on the diversity of the assistant professor pool in the short-term (through 2030; *Figure 5A*), and long-term (through 2080, *Figure 5B*). These strategies were: (i) increasing the size of the talent pool of PhDs from URM backgrounds (given the current transition rate), (ii) increasing the number of assistant professor positions available (given the current transition rate), or (iii) increasing the rate of transition of PhDs from URM backgrounds into the applicant pool of assistant professorships (with subsequent hiring). Unlike a "facsimile model" that would be designed to explicitly predict precise values of outcome metrics (in this case, precise numbers of PhD graduate and assistant professor ratios), our model is an "abstract model," meaning that simulations are intended to examine the qualitative behavior associated with hypothetical policy outcomes, assuming that the system continues to follow its historic behavior (*Gilbert, 2007*). Specifically, from 1980-2013, the number of URM PhD graduates grew at an exponential rate. Therefore, all model runs assumed continued exponential growth of URM PhD graduates (lower growth rates of PhD graduates did not change the qualitative behavior of our model's output).

In 2014, 5.8% of assistant professors in basic science departments were from URM

backgrounds. This level of representation is consistent with a transition rate of 0.25% of "other aspire" URM PhDs onto the market (*Figure 5—source data 1*). Given a 0.25% transition rate, short-term simulations (*Figure 5A*) showed that, increasing the size of URM talent pool or the numbers of assistant professor positions available were not sufficient to increase faculty diversity. Specifically, the model predicted that by 2030, 13.8% of biomedical PhDs would be URMs, but that only 5.9% of assistant professors would be URMs whether the number of assistant professor positions remained the same (*Figure 5Ai*) or grew by 100 positions annually beginning in 2015 (*Figure 5Aii*). Put another way, the model predicted that growing the URM PhD pool 53% above current levels (i.e. 13.8% v. the current 9%) would result in a less than 2% increase in the representation of URM assistant professors. Thus, in the presence of a low transition rate, the model predicted that increasing the size of the talent pool or the number of available positions would not lead to a significant increase in the representation of URM assistant professors through 2030.

Instead, the simulations predicted that increased diversity would result from increased transition of candidates onto the market and their subsequent hiring. Increasing the transition rate to 10% increased URM representation in the assistant professor pool to between 12.4–12.5% in 2030 given the same (*Figure 5Aiii*) or increased number of positions available (*Figure 5Aiv*). That is, increasing the transition rate increased URM faculty representation by more than two-fold above what would be predicted from simply increasing the number of URM scientists exponentially through 2030.

To test if there was a threshold above which the URM PhD talent pool was sufficient to result in increased faculty diversity, we conducted simulations through 2080 (with the caveat that as the time horizons extend unforeseen external factors are likely to arise that could attenuate predictive power). The model predicted if the URM PhD population continued to grow at an exponential rate through 2080, 73% of PhDs would be URMs (at which point these populations would no longer be underrepresented). However, in the presence of the 0.25% transition rate, the model predicted that in 2080 fewer than 10% of assistant professors would be URMs, no matter the number of positions available (*Figure 5Bi and Bii*). In contrast, the simulations predicted that if the transition rate increased to 10%, URM assistant professor

representation would be between 56.5-58.3% given the same (*Figure 5Biii*) or increased number of positions available (*Figure 5Biv*). Thus, these model simulations indicate that in the short and long-term, given the low transition rate, the size of the URM talent pool, and number of available positions in the overall market had a minimal impact on faculty diversity (even in the absence of labor market discrimination). Instead, increased faculty diversity resulted from ensuring the growing URM PhD and postdoctoral pools transitioned onto the job market and were hired (the impact of other transition rates are shown in *Figure 5—source data 1* and *2*). These simulations assume no racial bias in hiring; the presence of discrimination against URM scientists would attenuate any increases in faculty diversity.

## Discussion

Increasing faculty diversity in academic science departments has been a long-standing challenge and has received renewed attention in recent years (*Ginther et al., 2016*; *Ginther et al., 2011*; *Griffin, 2016*; *Duehren and Muluk, 2016*; *Myers et al., 2012*). Here, we used data from medical school basic science departments to highlight the impact of potential intervention strategies on the diversity of assistant professors. By illuminating some of the dynamics with respect to faculty diversity in medical schools, we aim to better understand diversity challenges in other segments of the scientific enterprise (including other university settings, industry, and government).

Although the dearth of URM faculty members in medical schools typically has been framed as a "pipeline" problem—i.e. a lack of available URM talent—our analysis shows that the rate of PhD production for scientists from URM backgrounds has increased significantly over the past 33 years, and at a faster rate than that of WR scientists. Despite this progress, there was no statistical linkage between the size of the pool of URM talent, and the number of URM assistant professors hired in basic science departments of medical schools (*Figure 2*; imputed values assuming 6-year turnover in assistant professor population; findings hold for longer periods of turnover). These findings suggest a decoupling of PhD production and faculty attainment in these environments for scientists from URM backgrounds. In contrast, WR assistant professor hiring numbers were more closely related to the total number of WR PhD graduates. Therefore,

broader changes in the biomedical academic labor market—i.e., more trainees than faculty positions, elongated pathways to independence, and declining research funding (*Alberts et al., 2014*; *McDowell et al., 2014*; *Larson and Ghaffarzadegan, 2014*)–are insufficient to explain differences in faculty attainment between postdoctoral scientists from URM and WR backgrounds.

Obtaining an assistant professorship position requires, at a minimum: (i) a position to be available; (ii) a candidate to be interested in and apply for the position; and (iii) the applicant to be favorably evaluated, offered the position and ultimately accept the position. Systematic data on the applications, evaluations, offers and acceptances for assistant professor candidates in any academic discipline are not typically made available (due to privacy and confidentiality laws), thus we were unable to include this information in the model. However, a recent study has shown that inequality and hierarchy characterize assistant professor hiring across disciplines, and that prestige of doctoral institution plays a major role in who is hired (*Clauset et al., 2015*). While 80% of Black and Hispanic science PhD graduates obtain their degrees from Carnegie classification research universities (high or very high research activity), this number is lower than the proportion of White and Asian PhD graduates from these institutions (90%) (*National Science Foundation, 2015b*). Thus, it is possible that part of the difference can be attributed to the nature of the assistant professor hiring process itself, which emphasizes training background. A more in-depth analysis of this aspect of faculty hiring the biomedical sciences remains a topic for future work.

Beyond institutional pedigree, previous work has shown that interest in assistant professor careers at research-intensive universities such as medical schools declines as training progresses (*Fuhrmann et al., 2011*; *Sauermann and Roach, 2012*), and these declines are larger for PhDs and postdocs from URM backgrounds relative to their WR counterparts (*Gibbs et al., 2015*; *Gibbs et al., 2014*). Importantly, differences in interest in these assistant professor positions between URM and WR scientists remained when controlling for first-author publication rate, advisor relationships, PhD training institution, research self-efficacy, and training experiences (*Gibbs et al., 2014*). This suggests that there are fundamental aspects of the environment, or nature of faculty work in research-intensive universities that cause otherwise equally qualified

URMs to differentially choose other career paths.

Indeed, prominent scientists from URM backgrounds have written about the unique experiences, challenges, and biases (implicit and explicit) faced while conducting science in these environments (*Colon Ramos and Quiñones-Hinojosa, 2016*; *Jarvis, 2015*). Further, there is an emerging body of literature on the distinct values that motivate many scientists from URM backgrounds to pursue scientific careers (e.g. giving back to or serving as a role model in their community of origin), the importance of congruence between personal values and career opportunities to fulfill them for scientists of all backgrounds, and the perception that faculty environments at research-intensive institutions such as medical schools may not enable sufficient engagement with the distinct values of URM scientists (*Gibbs et al., 2013*; *Estrada et al., 2011*; *Thoman et al., 2015*; *Smith et al., 2014*; *Powers et al., 2016*).

The modeling data we presented indicate that given current rates of transition from PhD to assistant professorship among URMs, the percentage of URM assistant professors in basic science departments of U.S. medical schools could remain below 10% in the short- and long-term (i.e. by 2080), even in the context of exponential growth of the URM PhD and postdoctoral pool and the absence of discrimination. Thus, faculty diversity efforts that rely primarily on enhancing rates of PhD graduates (i.e. "filling the pipeline") can only have their desired impact if they are coupled with efforts to get these candidates on the market and hired. This would require making faculty positions and work environments attractive and supportive to these scientists, ensuring the proper types of support (e.g. funding, mentorship and sponsorship) to allow URM postdocs to effectively progress to independence (*Valantine et al., 2016*), and ensuring institutional faculty recruitment, evaluation, and retention processes support scientists from all backgrounds (*Gasman, 2016*). Such efforts would have to take into account factors such as the broader landscape in which scientists from all backgrounds have greater career options (*Nature, 2014*), and the specific career development of women from URM backgrounds (*Gibbs et al., 2014*; *Gibbs et al., 2013*; *National Research Council, 2013*) who make up the majority of URM biomedical PhD graduates (*National Science Foundation, 2015b*).

While the challenge of achieving faculty diversity has been longstanding, with concerted and

targeted effort, the numeric realities of assistant professor hiring and turnover mean that higher diversity could be achieved relatively quickly among junior faculty. On average, the pool of assistant professors turns over every six years. The analysis presented here demonstrated that in recent years, assistant professor hiring has been relatively stable at around 1000 positions each year (or roughly 7 assistant professors per institution across all basic science departments annually). Thus, to achieve parity with the pool of PhD graduates (estimated to grow to 10% URM in 2016) would require hiring around 100 URM assistant professors annually at medical schools. Put another way, if roughly two-thirds of medical schools hired (and retained) just one faculty member from an URM background annually for the next six years, the system would reach parity with the PhD pool within one tenure cycle. While this would still not reflect the proportion of people from URM backgrounds in the overall population (currently greater than 30%), this would represent a meaningful first step to addressing the longstanding goal of enhancing scientific excellence by increasing faculty diversity.

## Materials and methods

### Data sources

Biomedical PhD attainment data were obtained from the National Science Foundation's Survey of Earned Doctorates (SED), an annual census of individuals receiving research doctorates from accredited U.S. institutions (*National Science Foundation, 2015a*), as compiled by the Federation of American Societies for Experimental Biology (FASEB) (*Garrison and Campbell, 2015*). FASEB tallies and publishes annually the number of PhDs granted in biomedical disciplines (including those who earned PhDs as a single degree or in combination with an MD) from 1980-2013. To calculate the total number of PhD graduates in the U.S., we added together the number of biomedical PhDs awarded to U.S. Citizens and permanent residents, temporary residents, and individuals with unknown citizenship. To calculate the number of PhD graduates from URM backgrounds, we added together the number of U.S. citizen and permanent resident PhD graduates who identified as one of the following: "Black/African-American (non-Hispanic/Latino)," "Hispanic/Latino," or "American Indian or Alaska Native" (*National Institutes of Health, 2015*). PhD

graduates from all non-URM backgrounds (White, Asian or Pacific Islander, "Other," "Unknown" and non-citizens) were called "well represented" (WR). These data are shown in *Figure 1—source data 1*.

To determine trends in representation of faculty in medical school basic science departments, we obtained faculty data from the AAMC Faculty Roster, 1980-2014 *Association of American Medical Colleges. 2015*. The AAMC Faculty Roster has collected comprehensive information on the characteristics of full-time faculty members at accredited allopathic U.S. medical schools since 1966. We focused specifically on assistant professors in basic science departments because on average 88% of the faculty members in these departments have earned PhDs (either as a single degree or in combination with and MD). For consistency, assistant professors who identified as "Black," "Hispanic/Latino," and "American Indian or Alaska Native" were considered URM. Assistant professors from all other backgrounds were considered WR. These data are also shown in *Figure 1—source data 1*.

### Assistant professor hiring trends

To calculate the aggregate number of assistant professors hired each year, we made two assumptions: (i) the length of time that an individual occupied the position of assistant professor was six years based on traditional academic promotion cycles (*Stanford University, 2016*; *Yale School of Medicine, 2014*), and similarly, (ii) one-sixth of the WR and URM assistant professors left the rank of assistant professor in 1980 (either to become associate professors, or pursue other career options). The numbers of assistant professors hired were then imputed based on real changes in the populations of URM and WR assistant professors each year. For example, if the assistant professor population changed from $n_1$ in year one to $n_2$ in year two, and the number of assistant professors leaving in year one was $l_1$, then the number of assistant professors hired in year two ($h_2$) equaled ($n_2$-$n_1$) + $l_1$. Thus, using real data showing that there were $n_1$=132 assistant professors from URM backgrounds in 1980, and $n_2$=129 in 1981, we assumed that one-sixth of the assistant professors left the rank in year 1980 ($l_1$=22), and thus $h_2$=19. These data are shown in *Figure 2—source data 1*.

Available evidence indicates that scientists who pursue faculty careers in the biomedical sciences remain in postdoctoral training for five or

more years (*National Institutes of Health, 2012a*). To estimate the pool of potential candidates for assistant professor positions, we assumed that PhD graduates (postdocs) remained in the pool of potential candidates for a total of five years, including the year in which they graduated (similar results were found when the length of time in the pool was four or six years). To calculate the pool of available candidates, we totaled the number of PhD graduates in the preceding five years, and subtracted the total number of assistant professors hired in medical school basic science departments in the preceding four years. For example, the pool of available candidates in the year 2010 equaled the sum of the PhD graduates from 2006–2010 less the candidates hired from 2006–2009. The percentage of the pool hired was derived by dividing the number of assistant professors hired each year by the size of the available pool. These data are shown in *Figure 2—source data 2*. These calculations were used to understand the current landscape and the connection between the available talent pool and faculty hiring, but were not used in the system dynamics model described below.

To determine the relationships between various constructs, we performed statistical analysis in R (version 3.2.2) (*R Core Team, 2015*). Specifically, we used a linear regression model to compare the growth of PhD graduates relative to the growth of assistant professors between URM and WR populations, where population relative to 1980 levels was the dependent variable, and independent variables included time (i.e. year), URM status (0= well-represented, 1=underrepresented minority), position (0=assistant professor, 1=PhD graduate), and the interaction between URM status and position. We also calculated Pearson's correlation coefficient between the size of the candidate pools and the number of assistant professors hired in each group. Finally, we examined how the proportion of the pool hired varied across time using a linear model with proportion assistant professors hired as the dependent variable and year as the independent variable, for scientists from URM and WR backgrounds. All figures were made in GraphPad Prism (version 6) and Adobe Illustrator (version 16.0.4).

### System dynamics model of assistant professor hiring

We used System Dynamics (SD) to create an abstract model (*Gilbert, 2007*) that could adequately explain macro-scale trends in the transition from the biomedical PhD pool into assistant professor positions in basic science departments at medical schools, and be used to predict the impacts of potential intervention strategies for increasing faculty diversity. We focused on entry into assistant professor positions since these positions turn over more rapidly than the overall faculty pool, and generally pull from recent PhD graduates and postdocs. All other areas of the workforce (i.e. faculty positions in other university contexts, research positions in industry or government, non-research positions occupied by PhDs) as well as longitudinal tracking of individual career transitions are beyond the scope of the model.

SD is a modeling framework that emphasizes the role of a system's structure on its ultimate behavior (*Homer and Hirsch, 2006*; *Meadows, 2008*), and has been used along with other approaches to model other aspects of the biomedical workforce and faculty hiring (*Ghaffarzadegan et al., 2014*; *Larson and Diaz, 2012*; *Ghaffarzadegan et al., 2015*). SD models implement these "loop-driven" dynamics through three basic types of elements: (1) stocks, which represent the accumulation of a quantity (e.g. assistant professors) and are represented by boxes; (2) flows, which represent the rate of change in a quantity (e.g. hiring rate for assistant professors), and are represented by hourglasses; and (3) variables, representing factors that can interact with stocks and flows in complex ways (e.g. number of assistant professors slots available), represented by words that have thin, blue causal arrows. In SD diagrams, clouds represent factors that are outside the system boundary, i.e., that extend beyond the range of the model. All modeling was done using Vensim PLE.

Our model extends beyond the standard "pipeline" conceptualization of faculty hiring (*Figure 3A*). There are three stocks accounting for progress from PhD graduation through assistant professor hiring: number of PhD graduates, number of candidates on the market (i.e., those PhD graduates who become postdoctoral scientists and pursue faculty careers), and number of assistant professors. Candidates on the market are hired into the stock of assistant professors at a rate equal to the total number of slots available. New assistant professors remain in the position for six years, at which point they leave the system boundary either through promotion or contract termination ("assistant professor tenure or leave"). Thus, the overarching structure of this model is that the number of assistant

professors hired is based on the number of slots available, and the number of candidates pursuing these positions.

We expanded upon this standard model in a number of ways (See *Figure 3B* for intermediate conceptual model, and *Figure 3C* for final model structure). All equations and initial parameter values are shown in *Appendix—tables 1* and *2*, and the final model is in the source code. Based on the analysis presented in *Figure 2*, and the distinct underlying patterns with respect to assistant professor position attainment after PhD completion, we separated the pathways for the career progression of URM and WR scientists. Further, for both URM and WR scientists, we assumed there was a fixed proportion of people who would pursue and enter assistant professor positions in medical school basic science departments (called "faculty aspire"), and all other PhDs would pursue careers in other sectors (called, "other aspire"; this can include research careers outside of academia, faculty careers in teaching-intensive environments, or careers away from bench research) (*Fuhrmann et al., 2011*; *Gibbs et al., 2015*; *Gibbs et al., 2013*).

The number of "faculty aspire" candidates is based upon the number of URM and WR assistant professors hired in the period of 1980-1997 (prior to the NIH budget doubling which lead a significant expansion in the number of biomedical PhDs). Without intervention, as the total number of PhD graduates grows, the pools of "URM faculty aspire," "URM other aspire," "WR faculty aspire," and "WR other aspire" are expected grow proportionally. The overall PhD graduate growth rate is represented as a variable termed "baseline PhD graduate growth rate." This is a piecewise linear function that mirrors the actual growth in the overall number of PhD graduates from 1980- 2013.

The model assumes all PhD graduates in the "faculty aspire" stock pursue assistant professorships in research-intensive environments. The remaining graduates pursue other careers and depart the system. Each year, the system attempts to fill all "slots available." Further, depending on market hiring conditions, 20% of the candidates on the market drop out of the market annually. That is, if the probability of being hired is relatively high (>20%) all candidates remain on the market that year. Otherwise, one-fifth of the available pool drops out of the system. Thus, the "half-life" of a candidate on the market who is not hired is 5 years (similar to the current period of postdoctoral training for

candidates pursuing faculty positions in research intensive environments [*National Institutes of Health, 2012a*]).

Candidates are chosen from the stocks of WR and URM candidates in proportion to their respective representations on the market. For example, if there are 100 slots available, and 25% of the candidates on the market are from URM backgrounds and 75% are from WR backgrounds, then 25 slots will be filled by URMs and 75 slots will be filled by WRs. Although the career development pathways for WR and URM students are separate, the total number of assistant professor slots available links them. If there are not enough candidates to fill all slots, these slots are removed from the system. Specifically, we posit that candidates in a specific group (URM or WR) are hired with likelihood proportional to their representation on the market. That is, the model posits no hiring bias.

Beyond the baseline student growth rate, "URM target growth rate" (*Figure 3C*) is a variable that represents the concerted effort of the scientific community (e.g. funders, institutions, scientific societies) to increase the representation of scientists from URM backgrounds in the PhD pool over and above what would occur with natural system growth. This growth rate increases exponentially throughout the duration of the model, matching the empirical trend of URM biomedical PhD population growth from 1980-2013. The "URM target growth rate" is conceptualized as the result of external intervention; thus, these additional URM scientists are initially added to the "URM other aspire" stock. For model simulations, the overall number of scientists in the "URM other aspire" stock is the greater of either the baseline growth rate, or the URM target growth rate. A proportion of URM PhD graduates who begin in the "other aspire" stock may then choose to pursue a faculty career. The "transition rate" is a free parameter that represents the proportion of URM "other aspire" graduates that choose to pursue faculty careers Because there has not been similar intervention to increase the number of scientists from WR backgrounds entering the PhD pool and faculty market, they do not have a transition rate. Thus, the total number of URM candidates on the market is equal to the number who would have entered the market in proportion to the overall growth of the system (i.e. "URM Faculty Aspire"), and the number who entered as a result of external intervention and then chose to pursue faculty careers (i.e. "URM Other Aspire" scientists who transition) minus those who have

dropped out of the market ("URM Market Dropout").

To determine the initial number of faculty aspire candidates for both URM and WR scientists ("Faculty Aspire $P_0$") we used the average number of assistant professors hired from 1980-1997, corresponding to the period before the NIH budget doubling, which drove significant growth of the biomedical enterprise (*Johnson, 2013*). The value of "URM Faculty Aspire $P_0$" is 34.4, and "WR Faculty Aspire $P_0$" is 567.7. The growth rates for each population (UR or WR) were then scaled to match empirical trends, resulting in numbers of PhD graduates. The patterns of growth differed between scientists from WR and URM backgrounds (piecewise linear and exponential respectively). Thus, for scientists from WR backgrounds, we used the difference between the 1980 population of PhD graduates and "WR Faculty Aspire" candidates to calculate the "WR other aspire $P_0$." For URM scientists, we fit an exponential function to URM student growth data and used the associated coefficient to derive the "URM other aspire $P_0$".

Finally, the number of assistant professor "slots available" is a piecewise function fit to data imputed from historical counts of assistant professors. Specifically, we made the simplifying assumption that there were a fixed number of slots available per year (i.e., flat hiring), with a step increase in 1998, when the NIH budget-doubling lead to an era of expansion in the biomedical sciences. Thus, the numbers of slots available from 1980-1997 (n=602.1 slots), and 1998-2013 (n=1064.7 slots) represent the average number of total assistant professors hired during these periods. The final model structure is shown in *Figure 3C* (the source code has model file).

### Limitations

There are a number of limitations to the analyses presented. Postdoctoral training has become an almost uniform prerequisite to obtaining a faculty position in biomedical research. There are no reliable estimates on the number of postdocs in the US (currently or historically), thus these data were not included in the model (*National Academy of Sciences NAoE, and Institute of Medicine, 2014*). However, available data suggest comparable postdoctoral transition rates across race/ethnicity for PhD graduates (*National Science Foundation, 2015b*). Further, we recognize that scientists from URM and WR groups are not all uniform (i.e. there are differences across race/ethnicity groups, and between them by dimensions including but not limited to socioeconomic background). However, the small numbers of scientists from any individual URM group on the faculty did not allow for further disaggregation and separate modeling of URM populations. Additionally, some scientists (from all racial-ethnic backgrounds) complete their training outside of the U.S., and then obtain faculty positions within the U.S. For example, a Black or Hispanic scientist who completed their training outside of the U.S., and then obtained a faculty position within the U.S. would be counted as a URM faculty member, even though they were not part of the URM PhD pool (which only includes U.S. Citizens and Permanent Residents who completed their training within the U.S.). There are no systematic data available on the numbers of scientists from either WR or URM racial-ethnic backgrounds who fit this category (*National Institutes of Health, 2012a*), thus these data were not included in the model. From the perspective of examining the transition from PhD to assistant professor in medical school basic science departments, a central focus of this work, the entry of scientists that identify as being part of an URM group into faculty positions but who were not trained in the U.S. would lead to an overestimation of the proportion of URM PhD graduates hired as assistant professors. Finally, between the period of 1980-2013, many universities adopted policies that allow for flexibility in the traditional six-year tenure clock (*American Association of University Professors, 2004*). Thus, our assumption of a fixed, six-year tenure clock may also lead to an overestimation of the number of assistant professors hired each year, but it is consistent with the standard timeframe used by number of leading institutions (*Stanford University, 2016*; *Yale School of Medicine, 2014*).

### Acknowledgements

The authors thank Andrew Miklos and Dorit Zuk for presubmission review and comments, and Malika Fair for helpful conversations.

**Kenneth D Gibbs  Jr** is in the Office of Program Planning, Analysis and Evaluation, National Institute of General Medical Sciences, Bethesda, United States

(iD) http://orcid.org/0000-0002-3532-5396

**Jacob Basson** is in the Office of Program Planning, Analysis and Evaluation, National Institute of General Medical Sciences, Bethesda, United States

ⓘ http://orcid.org/0000-0002-0521-3078

**Imam M Xierali** is in the Public Health and Diversity Initiative, Association of American Medical Colleges, Washington, United States

**David A Broniatowski** is in the Department of Engineering Management and Systems Engineering, The George Washington University, Washington, United States

ⓘ http://orcid.org/0000-0002-3302-9497

*Author contributions:* KDG, Conception and design, Acquisition of data, Analysis and interpretation of data, Drafting or revising the article; JB, Conducted analysis and wrote manuscript, Analysis and interpretation of data; IMX, Managed data collection and wrote manuscript, Acquisition of data; DAB, Conception and design, Analysis and interpretation of data, Drafting or revising the article

*Competing interests:* The authors declare that no competing interests exist.

## Additional files

### Supplementary files

• Source code 1. Vensim file containing the final system dynamics model of assistant professor hiring in basic science departments of medical schools.

### Funding

No external funding was received for this work.

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

## Appendix 1

# Model equations and parameter values

**Appendix 1—table 1.** Model formulation.

| Notation | Description | Formulation |
|---|---|---|
| UTG | URM Target Growth Rate | $\exp(t * C_{UTG})$ |
| $GR_{WR,Other}$ | WR Non-Faculty Student Growth Rate | $BSG * P_{0,WR,Other}$ |
| $GR_{WR,Faculty}$ | WR Faculty Student Growth Rate | $BSG * P_{0,WR,Faculty}$ |
| $GR_{URM,Other}$ | URM Non-Faculty Student Growth Rate | $BSG * P_{0,URM,Other}$ |
| $GR_{URM,Faculty}$ | URM Faculty Student Growth Rate | $MAX(BSG * P_{0,URM,Faculty}, UTG * P_{0,URM,Faculty})$ |
| $PHD_{WR,Other}$ | WR Non-Faculty PHD Graduates | $PHD_{WR,Other,0} + \int GR_{WR,Other} - DR_{WR,Other} dt$ |
| $PHD_{WR,Faculty}$ | WR Faculty Student PHD Graduates | $PHD_{WR,Faculty,0} + \int GR_{WR,Faculty} - MR_{WR,Faculty} dt$ |
| $PHD_{URM,Other}$ | URM Non-Faculty PHD Graduates | $PHD_{URM,Other,0} + \int GR_{URM,Other} - DR_{URM,Other} dt$ |
| $PHD_{URM,Faculty}$ | URM Faculty PHD Graduates | $PHD_{URM,Faculty,0} + \int GR_{URM,Faculty} - MR_{URM,Faculty} - TR_{URM,Faculty} dt$ |
| $DR_{WR,Other}$ | WR Non-Faculty Student Departure Rate | $PHD_{WR.Other}$ |
| $MR_{WR,Faculty}$ | WR Faculty Student Market Entrance Rate | $PHD_{WR.Faculty}$ |
| $DR_{URM,Other}$ | WR Non-Faculty Student Market Entrance Rate | $PHD_{WR.Other}$ |
| $MR_{URM,Faculty}$ | WR Faculty Student Market Transition Rate | $C_{UTR} * PHD_{URM.Faculty}$ |
| $TR_{URM,Faculty}$ | WR Faculty Student Departure Rate | $(1-C_{UTR}) * PHD_{URM.Faculty}$ |
| $PD_{WR}$ | WR Candidates on the Market (e.g., Postdocs) | $PD_{WR,0} + \int MR_{WR,Faculty} - HR_{WR} - DR_{WR,Faculty} dt$ |
| $PD_{URM}$ | URM Candidates on the Market (e.g., Postdocs) | $PD_{URM,0} + \int MR_{URM,Faculty} - HR_{URM} - DR_{URM,Faculty} dt$ |
| $\pi_{URM}$ | Proportion of URM candidates on the market | $PD_{URM}/(PD_{URM} + PD_{WR})$ |
| $HR_{WR}$ | Hiring rate of WR candidates | $MIN[PD_{WR}, S*(1- \pi_{URM})]$ |
| $HR_{URM}$ | Hiring rate of URM candidates | $MIN(PD_{URM}, S*\pi_{URM})$ |
| $DR_{WR,Faculty}$ | WR Faculty Student Departure Rate | $\begin{cases} 0 & if\ HR_{WR} < \frac{PD_{WR}}{5} \\ \frac{PD_{WR}}{5} & otherwise \end{cases}$ |
| $DR_{URM,Faculty}$ | URM Faculty Student Departure Rate | $\begin{cases} 0 & if\ HR_{URM} < \frac{PD_{URM}}{5} \\ \frac{PD_{URM}}{5} & otherwise \end{cases}$ |
| $AP_{WR}$ | WR Assistant Professors | $AP_{WR,0} + \int HR_{WR} - HR_{WR,t-6} dt$ |
| $AP_{URM}$ | URM Assistant Professors | $AP_{URM,0} + \int HR_{URM} - HR_{URM,t-6} dt$ |

Note: $HR_{.,t-6}$ denotes hiring rate of assistant professors delayed by six time steps (i.e., the length of a tenure cycle). For timesteps <7, $HR_{.,t-6}$ is calculated by amortization of the initial value of assistant professors $AP_{.,0}$.

**Appendix 1—table 2.** Parameters and exogenous variables.

| Notation | Description | Value | Source |
|---|---|---|---|
| $P_{0,URM,Faculty}$ | URM Faculty Student Growth Rate Multiplier | 34.44 | **AAMC Faculty Roster** (Imputed values, URM hiring 1980-1997) |
| $P_{0,URM,Other}$ | URM Non-Faculty Student Growth Rate Multiplier | 64 | **AAMC Faculty Roster** (Exponential fit of URM PhD graduate growth and imputed URM hiring 1980-1997) |
| $P_{0,WR,Faculty}$ | WR Faculty Student Growth Rate Multiplier | 566.67 | **AAMC Faculty Roster** (Imputed values, WR hiring 1980-1997) |
| $P_{0,WR,Other}$ | WR Non-Faculty Student Growth Rate Multiplier | 3500 | **AAMC Faculty Roster** (Linear fit of WR PhD graduate growth and imputed WR hiring 1980-1997) |
| $C_{UTG}$ | URM Target Growth Constant | 0.0728 | **FASEB** (Author estimation based on exponential fit to URM PhD graduation rate 1980-2013) |
| $PHD_{WR,Other,0}$ | Initial WR Non-Faculty PhD Graduates | 3570 | **FASEB** (Author estimation based on number of WR PhD graduates) |
| $PHD_{WR,Other,0}$ | Initial WR Faculty PhD Graduates | 438 | **FASEB** (Author estimation based on number of WR PhD graduates) |
| $PHD_{WR,Other,0}$ | Initial URM Non-Faculty PhD Graduates | 84.6 | **FASEB** (Author estimation based on number of URM PhD graduates) |
| $PHD_{WR,Other,0}$ | Initial URM Faculty PhD Graduates | 25.4 | **FASEB** (Author estimation based on number of URM PhD graduates) |
| $C_{UTR}$ | URM Transition Rate Constant | 0.0025 | **AAMC Faculty Roster** (Author estimation based on % URM Assistant Professor 2014) |
| $PD_{WR,0}$ | Initial WR Candidates on the Market | 511 | **AAMC Faculty Roster** (Imputed Hiring Value) |
| $PD_{URM,0}$ | Initial URM Candidates on the Market | 19 | **AAMC Faculty Roster** (Imputed Hiring Value) |
| $S$ | Faculty Slots Available per Year | Step function time series: $$S = \begin{cases} 601.11 & t \in [0,18) \\ 1063.67 & t>18 \end{cases}$$ | **AAMC Faculty Roster** (average of imputed hiring values: 1980-1997; 1998-2013) |
| $AP_{WR,0}$ | Initial WR Assistant Professors | 3246 | **AAMC Faculty Roster** |
| $AP_{URM,0}$ | Initial URM Assistant Professors | 132 | **AAMC Faculty Roster** |

