## [Decision Letter]

Thank you for submitting your article "Decoupling of the Minority PhD Talent Pool & Assistant Professor Hiring in Medical School Basic Science Departments" to *eLife* for consideration as a Feature Article. Your article has been reviewed by two peer reviewers and the *eLife* Features Editor, and this decision letter has been compiled to help you prepare a revised submission.

Summary:

This paper attempts to examine issues related to the hiring and retention of PhD underrepresented minorities (URM) in assistant professor positions at academic medical centers. It uses a system dynamics (SD) model to assess expectations of how many PhD URM individuals we should expect to see entering assistant professor positions. While others (Myers & Husbands-Fealing, 2012; Heggeness et al. 2016) have examined issues of representation along the 'pipeline,' this work expands on those efforts by examining specifically issues related to hiring and the flow and stock of assistant professors in academic medicine centers and, specifically, URM hiring. Additionally, the extention to an SD model is a unique analysis for this population and with this data.

Essential revisions:

1) This might be a discipline issue, but I find it confusing that in the body of the paper and in the figures, the data source is not cited! It is only cited in the back as an appendix of sorts. Data sources should be cited throughout the text and at the bottom of each figure.

2) Equation documentation should improve. In your appendix you should use more conventional approaches for model documentation than simple copy-paste of Vensim formula. Also, report all your parameters in a table with proper references (fine if it appears in the Appendix). For model documentation example, see other SD works such as Ghaffarzadegan, Hawley and Desai, 2014.

3) Definitions of subgroups need to be more precise. Instead of saying "including[…]" or "(i.e. white)," precise definitions should be clearly articulated within the paper of exactly how subgroups are defined.

4) One of the weaknesses of an SD model is that you are not able to account for individual agents' behavior, like in agent-based models. Lots of changes in the external environment (e.g. changes in funding streams, policies, and alternative opportunities) have the potential to influence an individual agent's actions in ways that are not necessarily linear and not accounted for in the model the authors present.

5) While the authors acknowledge the issue of postdocs in the supporting documentation, they do not even mention postdocs in the main paper. The reality is that hardly anyone transitions directly from PhD receipt to an assistant professor position, and the postdoc experience is extremely diverse. What if the differences the authors are seeing here are not a result of stalled entrance into the assistant professor position, but rather stalled entrance into a postdoc that will make the individual attractive for an assistant professor position. This dilemma and absence in their model must, at a minimum, be discussed in the body of the paper. Ideally, the authors would be able to incorporate some assumptions about the postdoc phase into their model.

6) The authors have no data on who is applying for assistant professor positions (as they acknowledge in the third paragraph of the Discussion). Therefore, there is no way to really claim, as they do, that if institutions increased their efforts to hire more URM, this would increase diversity in the assistant professor pool.

7) In the main model there is no drop-out from the pool of people in the market (that is, people stay in the market forever). I see that in your sensitivity analysis you report that you have conducted an analysis of effect of drop-out from the pool of people in the market. Very good, but I would argue that this should be in the main model. You simulate the model until 2080, and by then the whole population has retired and many have died! So, simply replace the results of your sensitivity analysis as the main analysis. I understand that this might not affect the results, but it is a better modeling practice.

8) The simulation period of 55 years in future is simply too long. And the assumption that 73% of PhD graduates will be URM feels unrealistic unless you provide evidence (if 73% of a population are minorities, then they are ORM: over-represented minorities!). I suggest the authors to simulate for the next 1-2 decades; there are many things that can happen until 2080 which your model cannot predict and are out of the boundary of your analysis.

9) One may argue that the reason "URM Faculty Aspire P0" is much smaller than the "WR Faculty Aspire P0" is that you are assuming that hiring in faculty positions is proportional to population of the pools. URM might be weaker in the market or be discriminated. From a modeling standpoint, your model has two degrees of freedom, if you assume there is no "weight" toward hiring WR relative to URM, you end up with much higher "WR Faculty Aspire P0" anyway. I think the best way, but difficult, is to provide some references for this argument (that there is less faculty aspiration among URM). The easier way is to clarify that you are aware of this assumption and discuss its implications in your policy recommendation. You may need to modify your language throughout the paper too.

---

## [Author Response]

[…]

*Essential revisions:*

*1) This might be a discipline issue, but I find it confusing that in the body of the paper and in the figures, the data source is not cited! It is only cited in the back as an appendix of sorts. Data sources should be cited throughout the text and at the bottom of each figure.*

We have added the following sentence to the main text and to Figure 1: “Data on the populations of PhD graduates and assistant professors in medical school basic science departments were obtained from the National Science Foundation Survey of Earned Doctorates (as compiled by the Federation of American Societies for Experimental Biology, and the AAMC Faculty Roster, respectively (please see Methods section for more information).”

*2) Equation documentation should improve. In your appendix you should use more conventional approaches for model documentation than simple copy-paste of Vensim formula. Also, report all your parameters in a table with proper references (fine if it appears in the Appendix). For model documentation example, see other SD works such as Ghaffarzadegan, Hawley and Desai, 2014.*

We have updated equation documentation consistent with the notation used by Ghaffarzadegan et al. (2014). This is now in appendix 1.

*3) Definitions of subgroups need to be more precise. Instead of saying "including[…]" or "(i.e. white)," precise definitions should be clearly articulated within the paper of exactly how subgroups are defined.*

We have endeavored to clarify these terms. The Introduction now reads:

“Scientists from certain underrepresented minority (URM) racial/ethnic backgrounds – specifically, African American/Black, Hispanic/Latin@, American Indian, and Alaska Native – receive 6% of NIH research project grants [Ginther, Kahn and Schaffer, 2016; Ginther et al., 2011; National Institutes of Health, 2012] despite having higher representation in the relevant labor market [Heggeness et al., 2016], and constituting 32% of the US population [National Institutes of Health, 2012].[…] This work focuses on three possible reasons for the low number of scientists from URM backgrounds in the professoriate relative to their peers from well-represented (WR) backgrounds (specifically, White, Asian, and all other non-URM groups) that are amenable to intervention by the scientific community”

The Methods now read:

“To calculate the number of PhD graduates from URM backgrounds, we added together the number of U.S. citizen and permanent resident PhD graduates who identified as one of the following: “Black/African-American (non-Hispanic/Latino),” “Hispanic/Latino,” or “American Indian or Alaska Native” [National Institutes of Health, 2015]. PhD graduates from all non-URM backgrounds (White, Asian or Pacific Islander, “Other,” “Unknown” and non-citizens) were called “well represented” (WR).”

We used definitions of “underrepresented minority” and “well-represented” groups consistent with the NIH definitions (http://grants.nih.gov/grants/guide/notice-files/NOT-OD-15-053.html), and conventions in the biomedical training and diversity literature:

http://journals.plos.org/plosone/article?id=10.1371/journal.pone.0114736

http://www.lifescied.org/content/15/3/ar41.long

http://www.lifescied.org/content/15/3/ar33.full

*4) One of the weaknesses of an SD model is that you are not able to account for individual agents' behavior, like in agent-based models. Lots of changes in the external environment (e.g. changes in funding streams, policies, and alternative opportunities) have the potential to influence an individual agent's actions in ways that are not necessarily linear and not accounted for in the model the authors present.*

In general, each modeling methodology has its strengths and weaknesses. System Dynamics (SD) models excel at capturing stocks, flows, nonlinear causal loops, and overall aggregate trends. The nature of our data is that they are aggregated and continuous. Thus, for the questions we aimed to address, specifically assessing the impact at an aggregate level of different intervention strategies on flows of PhDs from URM and WR backgrounds into faculty positions, SD modeling techniques are both adequate and preferred. This approach is standard in this domain; see, e.g., Larson and Diaz (examining the impact of retirement age on faculty hiring: https://www.ncbi.nlm.nih.gov/pmc/articles/PMC3737001/), and Ghaffarzadegan and colleagues (examining the impact of different strategies on systems level postdoc diversity: https://www.ncbi.nlm.nih.gov/pmc/articles/PMC4215734/). In order to further clarify this point, we have added the additional reference by Larson and Diaz throughout the paper to refer readers to other examples of how SD modeling can be used to address the types of questions posed here.

However, we agree with the reviewer that Agent-Based modeling (ABM) is better suited to modeling behavior at the level of the individual. In line with the reviewer’s comments, Bonabeau (2002) notes that ABMs excel when “Individual behavior is nonlinear and can be characterized by thresholds, if-then rules, or nonlinear coupling. Individual behavior exhibits memory, path-dependence, and hysteresis, non-markovian behavior, or temporal correlations, including learning and adaptation. Agent interactions are heterogeneous and can generate network effects.” In the event that we are able to gather data validating the extent to which these heterogeneities lead to systematic sources of error in our results, we hope to use ABM in future work.

*5) While the authors acknowledge the issue of postdocs in the supporting documentation, they do not even mention postdocs in the main paper. The reality is that hardly anyone transitions directly from PhD receipt to an assistant professor position, and the postdoc experience is extremely diverse. What if the differences the authors are seeing here are not a result of stalled entrance into the assistant professor position, but rather stalled entrance into a postdoc that will make the individual attractive for an assistant professor position. This dilemma and absence in their model must, at a minimum, be discussed in the body of the paper. Ideally, the authors would be able to incorporate some assumptions about the postdoc phase into their model.*

We thank the referee for pointing out our need for greater clarity, and have modified the main text to more clearly discuss the issue of postdocs. Specifically, we have:

Clarified that “candidates on the market” refers primarily to postdocs pursuing faculty positions in medical school basic science departments;

Highlighted the fact using the model that includes market drop out (see essential revision #7), the average half-life for “candidates on the market” who are not hired is five years, and that this consistent with the average length of time for postdoctoral study for candidates pursuing faculty positions in research-intensive environments (NIH Biomedical Research Workforce Report, 2012);

Added a brief discussion of focusing on postdoctoral transitions into the Discussion section.

We too recognize the importance of postdoctoral training in career development of biomedical PhDs (the lead author was previously on the Board of Directors for the National Postdoctoral Association). As has been noted by elsewhere (notably the 2012 NIH Biomedical Workforce Report, and the 2014 National Academies of Sciences report, “The Postdoctoral Experience Revisited”) data quality about postdocs in the United States remains very poor—estimates about their numbers range from 30,000 – 100,000. Thus, we did not feel there was adequate and rigorous data to include postdocs in the model.

With respect to question of stalled entrance into postdoc positions by URMs, available data from the NSF (specifically, table 8-3 of the “Women, Minorities and Persons with Disabilities” Report: https://www.nsf.gov/statistics/2015/nsf15311/tables/pdf/tab8-3-updated-2016-06.pdf; “Location and type of postgraduate activity for U.S. citizen and permanent resident S&E doctorate recipients with definite postgraduate plans, by ethnicity and race: 2014”) indicate that rates of postdoctoral transition are largely consistent across URM and WR status.

Specifically from 2004-2014 Hispanics were 53.4% of URM PhD graduates, African Americans are 43.3% of URM PhD graduates, and American Indian Alaska/Natives are 3.3% of URM PhD graduates. Using the NSF rates of postdoctoral study, we see that 41.5% of URM PhD graduates have definitive plans to progress to postdoctoral study post PhD (41.5% URM postdoc transition rate =0.534*(46.7% Hispanic postdoc rate) + 0.433*(35.7% African American postdoc rate) + 0.033 (33.3% American Indian/Alaska Native postdoc rate)).

Among WR groups, from 2004-2014, Whites were 55.6% of WR PhD graduates, Asians were 9.6% of WR graduates and “Other” ethnicities are 34.8% of WR PhD graduates. Using the NSF rates of postdoctoral study, we see that 42.6% of WR PhD graduates have definitive plans to progress to postdoctoral study (42.6% WR postdoc transition rate = 0.556*(41.7% White postdoc transition rate) + 0.096*(33.9% Asian postdoc transition rate) + 0.348*(44.7% Other” postdoc transition rate). We have added the following sentence to the manuscript when describing the model:

“Candidates on the market are composed primarily of the subset of postdoctoral scientists pursuing faculty careers in medical school basic science departments, and current evidence suggests that the rates of transition to postdoctoral training are comparable between URM and WR PhD graduates [Smith, 2015]”.

*6) The authors have no data on who is applying for assistant professor positions (as they acknowledge in the third paragraph of the Discussion). Therefore, there is no way to really claim, as they do, that if institutions increased their efforts to hire more URM, this would increase diversity in the assistant professor pool.*

We agree with the reviewer that systematic data regarding the demographics of faculty applicants would be ideal. However, these data are not available, which we acknowledge in the manuscript, “systematic data are not available on the demographics of faculty applicants.” However, it is established in the higher education and faculty development literature that increasing diversity in the applicant pool is a proven strategy to increase diversity in the professoriate (C. Turner, Diversifying the Faculty: A Guidebook for Search Committees, 2002; J. Moody, Faculty Diversity: Problems and Solutions, 2004; D. Smith, Diversity’s Promise for Higher Education: Making it Work, 2015).

These data are not likely to become available for many reasons, including confidentiality and legal restrictions (two examples of university HR policies regarding privacy are presented below):

http://hr.unc.edu/policies-procedures-systems/spa-employee-policies/personnel-information/personnel-records-and-confidentiality-of-personnel-information/

https://hr.uoregon.edu/records/classified-employee-records-and-data

Furthermore, Gibbs et al. 2014 demonstrated that URMs have lower interest in faculty positions at research-intensive universities (like medical schools) in comparison to their WR colleagues. In addition to the authors’ personal experiences (as one who works professionally in the area of workforce diversity, a very common reason given for the lack of faculty diversity is the lack of applicants), the lack of diversity in a recent applicant pool was emphasized and substantiated by a search committee chair at Harvard: http://www.nature.com/news/insider-s-view-of-faculty-search-kicks-off-discussion-online-1.19165). Therefore, we feel that it is appropriate to highlight the need to increase applications from URM scientists as a means of enhancing diversity. We have endeavored to clarify this logic in the manuscript and have added the additional references by Moody and Smith (Introduction, sixth paragraph).

*7) In the main model there is no drop-out from the pool of people in the market (that is, people stay in the market forever). I see that in your sensitivity analysis you report that you have conducted an analysis of effect of drop-out from the pool of people in the market. Very good, but I would argue that this should be in the main model. You simulate the model until 2080, and by then the whole population has retired and many have died! So, simply replace the results of your sensitivity analysis as the main analysis. I understand that this might not affect the results, but it is a better modeling practice.*

We have replaced the current model to incorporate market drop out and have updated the figures, figure legends, and text appropriately.

*8) The simulation period of 55 years in future is simply too long. And the assumption that 73% of PhD graduates will be URM feels unrealistic unless you provide evidence (if 73% of a population are minorities, then they are ORM: over-represented minorities!). I suggest the authors to simulate for the next 1-2 decades; there are many things that can happen until 2080 which your model cannot predict and are out of the boundary of your analysis.*

Per the reviewer’s suggestion, we have included a model run ending in 2030.

We have kept a longer model run in (through 2080) to illuminate for readers how given a “best case scenario” of a limitless URM talent pool and no discrimination, the current structure would not achieve increased faculty diversity unless there is also an increased transition rate. We have acknowledged in the text that there are many things that are likely to change that would impact long-term predictive capability of the model (see Discussion, second paragraph).

*9) One may argue that the reason "URM Faculty Aspire P0" is much smaller than the "WR Faculty Aspire P0" is that you are assuming that hiring in faculty positions is proportional to population of the pools. URM might be weaker in the market or be discriminated. From a modeling standpoint, your model has two degrees of freedom, if you assume there is no "weight" toward hiring WR relative to URM, you end up with much higher "WR Faculty Aspire P0" anyway. I think the best way, but difficult, is to provide some references for this argument (that there is less faculty aspiration among URM). The easier way is to clarify that you are aware of this assumption and discuss its implications in your policy recommendation. You may need to modify your language throughout the paper too.*

Two previous papers by the lead author provide evidence of lower interest in faculty careers at major research universities among scientists from URM backgrounds:

Gibbs et al., “Biomedical Science PhD Career Interest Patterns by Race/Ethnicity and Gender,” PLOS ONE 2014(PMID: 25493425; reference 24) demonstrated that at PhD completion, the URM men and URM women have lower interest than their WR counterparts in faculty positions at research-intensive universities, even when controlling for career pathway interest at PhD entry, first-author publication rate, faculty support, research self-efficacy, and graduate training experiences.

Gibbs et al., “Career Development of American Biomedical Postdocs,” CBE Life Sciences Education 2015(PMID: 26582238; reference 22), showed that in a cross section of American postdocs, URM women had the lowest interest in faculty positions at research-intensive universities of all groups, even when accounting for career pathway interest at PhD entry, first-author publication rate, faculty support, research self-efficacy, and graduate training experiences.

Therefore, there is evidence that apart from level of preparation, URMs have lower level of interest in faculty positions at research-intensive universities. We have highlighted these references more clearly in the manuscript (Introduction, fifth paragraph; subsection “Intervention Strategies to Increasing Assistant Professor Diversity”).

The P0 values were based on imputed hiring trends for scientists from URM and WR backgrounds from 1980-1997. That is, we assumed what happened historically with URM and WR faculty hiring would carry forward as the system grew. We have clarified this in the text.

We do make the conservative assumption of no discrimination. We do recognize that discrimination can remain a problem in academic hiring. However, we did not include this in the model because we were unaware of high quality, systemic evidence that could be used to quantify its impact. We have noted in the revised manuscript (subsection “Intervention Strategies to Increasing Assistant Professor Diversity”) that discrimination will further attenuate efforts to enhance faculty diversity. The goal of the model was to highlight to readers that even in the absence of discrimination, increasing faculty diversity will remain a challenge unless there is greater attention to enhancing post PhD (i.e. postdoctoral) transitions onto the market and their subsequent hiring. We thank the reviewer for helping us to clarify this point.